# Honest Students from Untrusted Teachers: Learning an Interpretable Question-Answering Pipeline from a Pretrained Language Model

## Abstract

Explainable question answering systems should produce not only accurate answers but also rationales that justify their reasoning and allow humans to check their work. But what sorts of rationales are useful and how can we train systems to produce them? We propose a new style of rationale for open-book question answering, called *markup-and-mask*, which combines aspects of extractive and free-text explanations. In the markup phase, the passage is augmented with free-text markup that enables each sentence to stand on its own outside the discourse context. In the masking phase, a sub-span of the marked-up passage is selected. To train a system to produce markup-and-mask rationales without annotations, we leverage in-context learning. Specifically, we generate silver annotated data by sending a series of prompts to a frozen pretrained language model, which acts as a teacher. We then fine-tune a smaller student model by training on the subset of rationales that led to correct answers. The student is "honest" in the sense that it is a pipeline: the rationale acts as a bottleneck between the passage and the answer, while the "untrusted" teacher operates under no such constraints. Thus, we offer a new way to build trustworthy pipeline systems from a combination of end-task annotations and frozen pretrained language models.

## 1   Introduction

To be trustworthy and useful, a question answerer should be able to explain its reasoning and offer evidence. In open-book question answering, such explanations often take the form of rationale *masks*, which are subsets of tokens from the original passage [18]. However, a challenge for mask-based rationales is that subspans of the original passage are not meant to be read alone: coherent texts contain anaphora, ellipsis, and other cohesion-building elements that limit the interpretability of individual subspans when extracted from the discourse [13]. An example is shown in Figure 1, in which the key sentence mentions the answer only through the nominal *the grieving goddess*. A sufficient rationale for this answer would have to include an additional sentence introducing the entity *Astarte* and binding it to the nominal in the sentence that describes the key event.

Despite their limitations, extractive rationales have an important advantage over free-text explanations: they are directly linked to the original passage, making it easy for human readers to assess the reliability of the evidence for themselves. In this paper, we present a new style of explanation, called **markup-and-mask**, which preserves the attributability of extractive rationales while overcoming the problems created by extracting propositions from the discourse in which they were written. The key

- **Question:** What is the name of the person who revived Eshmun?
- **Passage:** ... Eshmun, a young man from Beirut, was hunting in the woods when Astarte saw him [Eshmun] and was stricken by his [Eshmun] beauty. ... The grieving goddess [Astarte] revived Eshmun and transported him [Eshmun] to the heavens where she [Astarte] made him [Eshmun] into a god of heaven. ...
- **Answer:** Astarte.

Figure 1: An example from QuoRef [8] with the generated rationale shown in dark text. The markup, shown in square brackets, makes it possible to find a more concise rationale than could be extracted from the original passage.

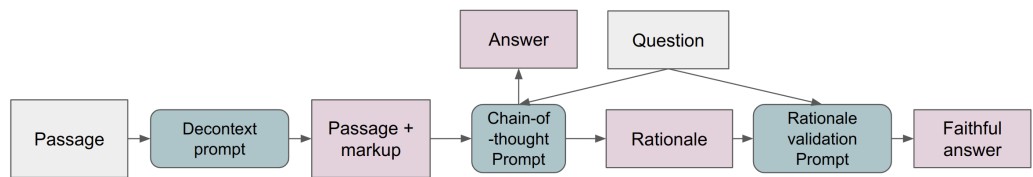

Figure 2: Schematic of the prompt chain used to produce silver data to fine-tune the honest student. At the decontextualization stage, one prompt is applied per sentence in the passage in sequence; the remaining stages use exactly one prompt each.

idea is that discourse context is made explicit in free-text markup and then rationales are extracted from the marked-up passages.

Rather than annotating markup-and-mask rationales manually, we present a new training method that leverages the in-context learning capability of large pretrained language models (Figure 2). First, we prompt a frozen language model to produce markup that sequentially decontextualizes each sentence in each passage in the training set. Next, we prompt the same language model to produce answers and chain-of-thought rationales from the decontextualized passage. Finally, we check that the rationale supports the answer by prompting the language model again, this time replacing the full passage with the rationale. When the answer approximately matches the ground truth, we add the rationale and markup to a silver training set. These silver annotations are used to train an "honest student" that is constrained to follow a pipeline: first generate question-neutral markup, then select a question-based rationale, and finally produce an answer using the rationale and not the passage.

Evaluation shows a number of favorable properties to this approach: (1) unlike other masking-based methods, accuracy on SQuAD is nearly as good as that of an end-to-end system; (2) on QuoRef, markup significantly increases accuracy; (3) answers that can be validated by a rationale are much more likely to be correct (+20 $F_1$); (4) rationales usually entail the answers; (5) despite having access to only five human-annotated examples of decontextualizing markup, the student model produces markup that is more accurate than a system that was fine-tuned on 11,290 gold-labeled training examples. The student models outperform their teacher on all three of our key metrics — overall accuracy, entailment rate of rationales, and accuracy of decontextualizing markup — highlighting the positive impact of distillation from pretrained language models.

To summarize the contributions of this paper:

- We propose markup-and-mask rationales for open-book question answering, which preserve a direct link to the original evidence text but use markup to incorporate non-local information.

- We show that it is possible to train models to produce markup-and-mask rationales without explicit supervision, by leveraging the capabilities of a pretrained language model.

- We present a general strategy for using pretrained language models to help supervise interpretable pipeline systems in which annotations are available for only the end task.

- We empirically validate the proposed approach, showing that the resulting rationales: (1) support accurate question answering; (2) help quantify predictive uncertainty; (3) are more likely to entail the predicted answers than "chain-of-thought" rationales produced alongside the answer; and (4) accurately match human-written decontextualizations.

## 2 Generating Markup-and-Mask Annotations

Our goal is to fine-tune a student model to produce markup-and-mask rationales. Lacking labeled examples, we obtain silver annotations by applying three distinct prompting patterns to the pretrained language model PaLM [5] (540-billion parameter version), which we refer to as the *teacher model*. Each prompt combines passages and questions from open-book question answering datasets, along with the outputs of previous prompts, in an approach that has been called *prompt chaining* [28]. There are three steps to the silver annotation process: (1) decontextualization; (2) chain-of-thought question answering; (3) rationale validation. The prompt chain is shown in Figure 2.

**Decontextualization.**   The goal of the decontextualization step is to add free-text markup of the style shown in fig. 1. Decontextualization examples are linearized as `Context:  ... Passage: ... Rewrite:`, with the language model prompted to complete the rewrite. An example is shown in Figure 5. We use a hand-crafted prompt with five examples, shown in appendix A. We proceed incrementally through the document, decontextualizing each sentence using the previous $k$ decontextualized sentences as context. This enables information to propagate through the document.

The capabilities and limitations of this approach are highlighted in Figure 6, which shows some typical outputs. The markup resolves pronominal references *she* and *her* and the nominal references *this painting* and *this phenomenon*. Perhaps most impressively, the elliptical expression *despite this* is decontextualized with the markup *[the fact that nudes were extremely rare. . . ]*. However, by the end of the document, we have lost track of the first name of the artist, so that *the artist* is decontextualized as only *[Velázquez]*, rather than with the full name. Future work may address this issue by exploring more sophisticated strategies than simple autoregressive decontextualization.

**Chain-of-thought question answering.**   In chain-of-thought prompting, the language model is asked to first generate a rationale before producing an answer [27]. For open-book question answering, we take the rationale to be a sentence that is extracted from the passage and which contains the answer, as shown in Figure 7. We construct question-specific few-shot prompts by concatenating several exemplars in which a question, passage, rationale, and answer are shown, before providing the question and passage for the instance to be predicted. The exemplars are drawn from the training set, selecting questions with the highest BM25 similarity to the target question [24]. Exemplars are added until we reach a limit of 1024 sentencepiece tokens in the prompt [17]; for the QuoRef dataset, this amounts to two or three exemplars in most cases.

To generate the rationales in the exemplars, we enumerate all sentences in the passage that contains an exact match to the answer and select the one with the highest BM25 similarity to the exemplar's question. Each sentence is considered in both its original surface form and with decontextualizing markup. If no sentence contains an exact match to the answer, then the question is not included as an exemplar. However, prompts are constructed for all training set examples, even when no rationale can be extracted using this heuristic.

**Rationale validation.**   Finally, to validate the rationales that were generated in the chain-of-thought stage, we perform a final validation stage in which the teacher model must answer questions based only on the generated rationales. As in the previous stage, we include each training set example and construct in-prompt exemplars by BM25 similarity to other questions in the training set. Because this stage does not include full passages, we can fit many more exemplars while remaining under the budget of 1024 tokens, on the order of 20 per prompt. The resulting "faithful answers" are then used to filter the fine-tuning data that is exposed to the student model.

## 3 Training the Student Model

The prompt chain described in Section 2 produces markup-and-mask rationales and uses them to answer questions. However, there are two main reasons to distill this teacher model into a smaller "honest student." The first reason is efficiency: the prompt chain requires several calls to the large language model; because it is more specialized, the student model can potentially be smaller. The second reason is accuracy: in the teacher model, the training set is used only for in-context learning, with only a few examples per prompt; fine-tuning can make use of more gold answers, in combination with silver rationales.

To fine-tune the student model, we use as training data the gold answers and the rationales produced by the teacher model. Because our goal is to train an *honest* student, we implement the student model as a pipeline: it must first produce the decontextualizing markup without seeing the question, then generate a rationale from the passage (conditioned on the question and the marked-up passage), and finally produce an answer (conditioned on the question and the generated rationale). Critically, the student does not consider the full passage when generating the answer. Each step of the pipeline is implemented as a text-to-text model using the t5x library [23], and the steps are trained in a single multi-task model. The specific tasks for the student model are:

**Decontextualizing markup.** As in the teacher model, decontextualization is performed autoregressively, with one training example per sentence. The target output is the markup produced by the teacher model.

**Span selection.** The input to the span selection task is a concatenation of the question and the decontextualized passage, and the target output is the rationale generated by the teacher in the chain-of-thought QA step. At training time the decontextualized passages are from the teacher; at prediction time they are from the decontextualizing markup step in the student pipeline.

**Rationale-based reading comprehension.** At training time, the input is a concatenation of the question and the teacher model's rationale; the target output is the gold answer. At prediction time, the input includes the rationale produced by the span selection step in the student pipeline.

**End-to-end reading comprehension.** For comparison, we also train an end-to-end reading comprehension task, in which the input is a concatenation of the question and the full passage. The target output is the gold answer and no rationale is produced.

The decontextualization task aligns closely to the decontextualization *prompt*, but the student model is trained by fine-tuning while the teacher model relies only on in-context learning. Unlike the chain-of-thought prompt described in Section 2, the span selection task does not produce an answer; the rationale-based reading comprehension task is conceptually similar to the rationale validation prompt, but again, the student model uses fine-tuning rather than in-context learning. To build a cleaner silver training set, we train only on the rationales that led to approximately correct answers at both the chain-of-thought stage (using the entire passage) and the validation stage (using the rationale alone). Specifically, we score the generated answers at both stages, and exclude examples for which either answer has an $F_1 < 0.5$.

## 4   Evaluations

We evaluate on two datasets: QuoRef [8] and the version of SQuAD [22] from the MRQA shared task [11]. For each dataset, we run PaLM on the training data to produce silver annotations of the markup-and-mask rationales, as described above. The decontextualization step is autoregressive, in the sense that the decontextualization for sentence $t$ is part of the prompt for decontextualizing sentence $t + 1$. This makes it difficult to use the more efficient bulk inference procedure that we apply in the other parts of the prompt chain. For this reason, we use only a fraction of the SQuAD training data (12000 questions). We then use PaLM's output as annotations to fine-tune multitask sequence-to-sequence models built on pretrained mT5 backbones [30]. The results that follow are based on the mT5-XXL backbone. Comparisons across model scales are shown in Figure 3.

### 4.1   Accuracy

Table 1 shows the overall performance of the student model, an end-to-end equivalent, and a masking-only ablation. On the SQuAD dataset, performance is similar across all model variants, showing that it is possible to derive causal rationales for SQuAD answers with only a minimal impact on accuracy. In contrast, prior work has found that previous unsupervised techniques for constructing rationales [21, 12] decreased performance by 10-20 $F_1$ on SQuAD [3]. The pipeline method suffers a significant reduction in accuracy on QuoRef, which, as discussed below, is particularly resistant to rationale-based approaches. However, this is mitigated by the use of decontextualizing markup, reducing the gap between the end-to-end predictor and the mask-based rationales by almost half.

**Selective prediction.**   The availability of a step-by-step explanation can serve as a coarse form of calibration: examples for which explanations are available may be more likely to be accurately

|                            | SQuAD       | QuoRef      |
|----------------------------|-------------|-------------|
| End-to-end (mT5-XXL)       | 83.2 / 92.8 | 80.4 / 85.8 |
| **Honest students (mT5-XXL)** |          |             |
| Markup+mask                | 82.2 / 91.7 | 68.2 / 74.5 |
| Mask-only                  | 82.2 / 91.7 | 51.9 / 58.9 |
| **Teachers (540B)**        |             |             |
| PaLM in-context            | 73.7 / 86.2 | 57.9 / 66.7 |
| PaLM in-context (+markup)  | 71.9 / 84.9 | 50.6 / 60.0 |

Table 1: Overall exact match / $F_1$ on open-book question answering. The *end-to-end* system predicts the answer directly from the passage; the *markup+mask* system predicts the answer from a rationale that includes both masking and markup; the *mask-only* system uses a rationale based only on masking the original unmarked text; *PaLM in-context* refers to the teacher model, which uses in-context learning only.

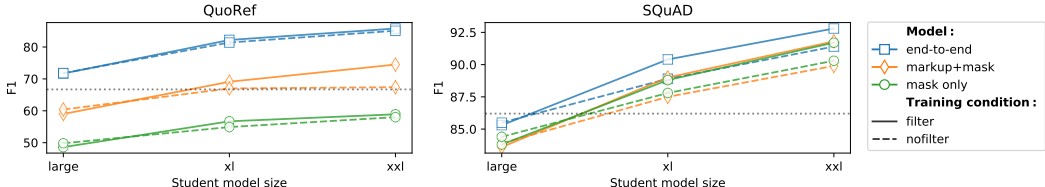

Figure 3: Overall $F_1$ results by student model size, for each configuration. The teacher model $F_1$ is shown with the dotted horizontal line.

predicted. To test this idea, we compare accuracy on examples where the end-to-end model and the rationale-based pipeline agree and disagree. As shown in Table 5, rationalizable answers are significantly more accurate. The $F_1$ for rationalizable answers is more than 20 points higher than for non-rationalizable answers on both datasets, and the gap in exact match is even larger. Furthermore, most answers are rationalizable in this way. The markup-and-mask rationales play an important role in selective prediction on the QuoRef dataset, where they increase the fraction of rationalizable answers from 58% to 74%, while enlarging the $F_1$ gap from 13.0 to 22.1. However, on the QuoRef dataset, a better coverage-accuracy tradeoff can be obtained by thresholding on the predictive probability of the end-to-end model; on SQuAD, the tradeoff is almost identical.

## 4.2 Rationales

To test how often rationales are consistent with the answers, we apply natural language inference (NLI). Specifically, we ask a strong NLI system whether the rationale entails the linearization, "The answer to "[question]" is "[predicted-answer]"". This style of evaluation has been applied to other tasks involving factual consistency, such as summarization and fact verification [14]. We use a very similar NLI system, trained by fine-tuning t5-XXL on multiple NLI datasets (MNLI, SNLI, FEVER, PAWS, SciTail, and VitaminC). As shown in Figure 4, the rationales produced by the pipeline student models are significantly more consistent than the chain-of-thought rationales produced by the teacher model, justifying the "honest student" moniker. On the QuoRef dataset, 64% of the rationales produced by the student model (with markup) entail that model's predicted answers, versus 47% for the teacher model with markup, and 36% without. On the SQuAD dataset, the student model achieves 81% consistency, versus 76% for the teacher model (75.5% without markup). The markup also improves the consistency of the student model by 26% on QuoRef and 1% on SQuAD. It is particularly notable that markup improves the entailment rate despite the fact that the NLI system is trained on data that does not contain any markup.

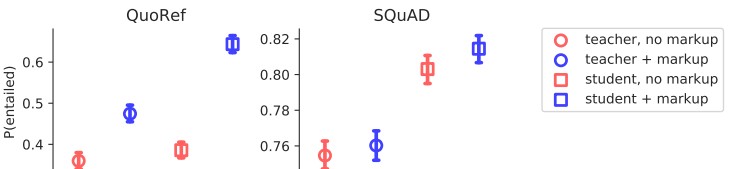

Figure 4: Consistency of rationales, as measured by the frequency with which the rationale entails a linearization of the question and the predicted answer.

|  | SQuAD | QuoRef |
|---|---|---|
| Passage length | 178.9 | 491.7 |
| Rationale length | 39.9 | 62.1 |
| Markups per passage | 4.8 | 31.6 |
| Mean tokens per markup | 5.6 | 5.3 |
| Median tokens per markup | 4.0 | 4.0 |
| % Extractive rationales | 90.6 | 92.3 |
| % Passages with faithful markup | 85.4 | 73.4 |
| % Sentences with faithful markup | 96.4 | 96.7 |

Table 2: Passage-level statistics of the rationales produced by the XXL-based models. Passage length and rationale length are computed in number of SentencePiece tokens. For more details on the other statistics, see Sections 4.2 and 4.3.

**Extractiveness and compression.** A rationale is deemed *extractive* when it appears as a contiguous substring in the marked-up passage, case-insensitive and not including punctuation characters or whitespace. Extractiveness is desirable because it means that the rationales are directly grounded in the passage, similar to the notion of "verified quotes" proposed by [20]. In QuoRef, the student model rationales were extractive for 92.3% of passages; in SQuAD, 90.6%. These rationales yielded 7.9x compression in QuoRef and 4.5x compression in SQuAD. Basic statistics of the markup and rationales are shown in Table 2.

### 4.3 Decontextualizing markup

To measure the accuracy of the decontextualizing markup, we apply the prompt-based teacher and the fine-tuned student models to a manually decontextualized dataset, in which references are replaced inline rather than annotated with markup [4]. On the SARI-Add metric [29], the teacher achieves $F = 0.32, P = 0.62, R = 0.21$ and the XXL-scale student (trained on QuoRef) achieves $F = 0.33, P = 0.67, R = 0.22$ (see Table 4 for details). These exceed the reported results for a T5-base model that was fine-tuned on 11,290 in-domain examples of the decontextualization task ($F = 0.29, P = 0.67, R = 0.19$); the state-of-the-art fine-tuned XXL-scale model achieves $F = 0.42, P = 0.72, R = 0.30$. This shows that it is possible to learn to perform the task reasonably well from just five labeled examples, and that distillation improves performance further. Our models produce a different style of decontextualization from the test data, so it is possible that these results could be further improved.

**Fidelity.** Because markup is a free-text generation task, it may not be *faithful*: the removal of markup may not yield a passage that is alphanumerically identical to the original passage (case-insensitive). The student model's decontextualizating markup achieves similar levels of fidelity to those of the prompt-based teacher. For more than 96% of sentences in the QuoRef dataset, the decontextualization phase leaves the original text unaffected, as intended; in 73% of passages, all markup was faithful. In the SQuAD dataset, the decontextualization was faithful in 96% of sentences and in 85% of full passages. The difference at the passage level is due to mainly the greater length of the QuoRef passages (see Table 2).

The teacher model markup was slightly less faithful: on both the SQuAD and QuoRef datasets, approximately 94% of the teacher model's sentence decontextualizations were faithful. This indicates that the language model can learn the format of the markup task from the five in-context examples. Most of the errors were minor, such as omission of sentence-final punctuation and the erroneous movement of text from the original into markup, e.g. *As a schoolboy Saint-Saëns was outstanding → As a schoolboy [Charles-Camille Saint-Saëns] was outstanding*. More serious errors, such as incorrectly-formatted markup and deletion of significant original content, occurred very rarely.

**Amount of markup.** On the QuoRef dataset, the decontextualization model added 2.0 markup spans per sentence, with an average length of 5.3 SentencePiece tokens per span (31.6 per document). This almost exactly matches the behavior of the teacher model, which added 2.1 spans, with 5.8 SentencePiece tokens per span (median=4). On the SQuAD dataset, there were fewer opportunities for decontextualization: the teacher model added 0.9 markup spans per sentence, with 6.1 tokens per span. The student model also added 0.9 spans per sentence (4.8 per document), with 5.6 tokens per span (median=4).

# 5   Related Work

Philosophically, the honest student is motivated by the goal of increasing the *warranted trust* in question anwering systems [15], by building an architecture in which the rationales (1) meaningfully constrain the predicted answer, and (2) can easily be checked by users.

**Rationales for question answering.**   Rationales are typically defined as masks on the input passage [18], with the goal of finding the minimal rationale that is sufficient to identify the ground truth label [9]. Such masks can be learned from human annotations [31, 20] or from unsupervised objectives such as information bottleneck [21]. We depart from fully extractive rationales by adding decontextualizing markup, unlike prior work in which decontextualization is performed inline [4], obscuring the relationship to the original text. This markup often indicates coreference relationships. Prior work has used human annotations to capture coreference in question answering [10]. We show that similar functionality can be obtained without human annotations, through the combination of in-context learning and end-task supervision.

**Reasoning chains in language models.**   In the past year, a number of papers have explored the ability of large language models to "show their work." In chain-of-thought and least-to-most prompting, the model is prompted to produce an explanation alongside its answer, with questions focusing on arithmetic and commonsense reasoning [16, 27, 32]. In all of these papers, the purpose of the explanations is not necessarily to make the model more trustworthy, but rather, to make the answer more accurate. Concurrent work uses chain-of-thought prompting in a student-teacher setup, similar to our architecture [25]. Unlike in our approach, the chain-of-thought is ignored and the focus is exclusively on the end-task accuracy of the student. Another key difference from prior work on chain-of-thought prompting is that our ultimate goal is to build an *honest* student model, whose rationales accurately describe the passage and the predicted answer [6].

Another line of work has focused on training language models to perform reasoning by fine-tuning on gold reasoning traces [2, 6, 7, 26]. In contrast, our work does not rely on annotations of reasoning traces: our student model learns to perform accurate multi-step inferences by relying on the combination of few-shot in-context learning and filtering on the performance of the end-task. In this way, our approach is more similar to [16], in which the model is fine-tuned to rationalize its predictions by "bootstrapping" from a small number of labeled examples. We provide a conceptually simpler approach that trains a student model by leveraging the pretrained capabilities of a large language model, eliminating the need for even a small seed set of labeled examples (except for the decontextualization step, which includes five labeled sentences), and using standard fine-tuning rather than a more complex iterative procedure with a dynamic training set.

# 6   Discussion

We show how to train an *honest student* to produce markup-and-mask rationales for open-book question answering. The approach has three key properties: (1) the rationales are more *expressive* than traditional masks because they include free-text markup to enable each sentence to stand on its own; (2) the rationales are *faithful* because the student model must first produce the rationale and then discard all other information from the passage when answering the question; (3) the rationale-generation system is *unsupervised*, training on silver data created by prompting a large language model. These properties suggest a general methodology for a new generation of pipeline systems, which could offer the benefits of interpretability and controllability while limiting annotation cost and achieving the expressivity of natural language. In future work we will explore the capability of the teacher model to support even more expressive reasoning patterns, through richer prompt chains.

**Limitations.**   A number of limitations are highlighted by the error analysis in Appendix D. More generally, we have assumed that answers can be rationalized by a contiguous span of the passage, after applying query-independent markup. This explains the lower performance of the pipelined methods on QuoRef, which contains questions that are hard to answer from any single sentence, even with query-independent markup. Another limitation is that markup is provided in a single forward pass, making it impossible to handle cataphoric references — for example, when an individual's name is revealed only at the end of a passage.

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

## A Prompts

During decontextualization, the language model must be queried for every sentence in the dataset. For this reason, and because results were promising from the first exploratory prompts, we did not

consider many alternative prompts. The prompt was written to include a few types of decontextual-
ization, including references to people, locations, times, and events, as well as cases in which the
decontextualizing information was not present in the context.

```
Instructions: rewrite each Passage using the Context.

Context: Lisa loves to play practical jokes.
Passage: But sometimes she goes too far.
Rewrite: But sometimes she [Lisa] goes too far.
Context: Bruce Lee is buried in Seattle.
Passage: But some of his biggest fans don't know he is from there.
Rewrite: But some some his [Bruce Lee] biggest fans don't know he
↪  [Bruce Lee] is from there [Seattle].
Context: The Super Bowl XLI halftime show took place on February 4,
↪  2007.
Passage: It was headlined by Prince.
Rewrite: It [The Super Bowl XLI halftime show] was headlined by
↪  Prince.
Context: Many years later as he faced the firing squad, Colonel
↪  Aureliano Buendia was to remember that distant afternoon when
↪  his father took him to discover ice.
Passage: At that time Macondo was a village of twenty adobe houses.
Rewrite: At that time [when his father took him to discover ice]
↪  Macondo was a village of twenty adobe houses.
Context: Ursula lost her patience.
Passage: If you have to go crazy, please go crazy all by yourself!
↪  she shouted.
Rewrite: If you [UNKNOWN] have to go crazy, please go crazy all by
↪  yourself [UNKNOWN]! she [Ursula] shouted.
```

These exemplars are then combined with individual sentences and contexts, as shown in Figure 5.

```
Instructions: rewrite each passage using the context.

[in-context exemplars]

Context: The Rokeby Venus (also known as The Toilet of Venus, Venus
↪  at her Mirror, Venus and Cupid, or La Venus del espejo) is a
↪  painting by Diego Velázquez, the leading artist of the Spanish
↪  Golden Age.
Passage: Completed between 1647 and 1651, and probably painted
↪  during the artist's visit to Italy, the work depicts the
↪  goddess Venus in a sensual pose, lying on a bed and looking
↪  into a mirror held by the Roman god of physical love, her son
↪  Cupid.
Rewrite: Completed between 1647 and 1651, and probably painted
↪  during the artist's [Diego Velázquez] visit to Italy, the work
↪  [The Rokeby Venus] depicts the goddess Venus in a sensual pose,
↪  lying on a bed and looking into a mirror held by the Roman god
↪  of physical love, her son Cupid.
```

Figure 5: Linearization of a single decontextualization example. The text after "Rewrite: " is
the model output. For subsequent sentences, the context includes the decontextualized sentences,
enabling information to propagate through the entire document.

An example prompt for chain-of-thought QA is shown in Figure 7. As described above, the in-context
exemplars are selected from the training set dynamically, based on similarity to the question.

```
She [Venus] is often described as looking at herself on the mirror,
↪   although this is physically impossible since viewers can see
↪   her [Venus] face reflected in their direction. This phenomenon
↪   [Venus gazing at herself on the mirror] is known as the Venus
↪   effect.
...
Nudes were extremely rare in seventeenth-century Spanish art, which
↪   was policed actively by members of the Spanish Inquisition.
↪   Despite this [the fact that nudes were extremely rare in
↪   seventeenth- century Spanish art, which was policed actively by
↪   members of the Spanish Inquisition], nudes by foreign artists
↪   were keenly collected by the court circle, and this painting
↪   [The Rokeby Venus] was hung in the houses of Spanish courtiers
↪   until 1813, when it was brought to England to hang in Rokeby
↪   Park, Yorkshire.
...
The painting [The Rokeby Venus] is believed to have been executed
↪   during one of Velázquez's [the artist] visits to Rome, and
↪   Prater has observed that in Rome the artist [Velázquez] "did
↪   indeed lead a life of considerable personal liberty..."
```

Figure 6: Example of output from the decontextualization prompt, applied to the Wikipedia page
`https://en.wikipedia.org/wiki/Rokeby_Venus`

```
Use each passage to answer the question, and cite the most relevant
↪   sentence as an explanation.

[in-context exemplars]

Question: What is the name of the person who revived Eshmun?
Passage: The myth of Eshmun was related by the sixth century Syrian
↪   Neoplatonist philosopher Damascius ...
Explanation: The grieving goddess [Astarte] revived Eshmun and
↪   transported him [Eshmun] to the heavens where she [Astarte]
↪   made him [Eshmun] into a god of heaven.
Answer: Astarte.
```

Figure 7: An example prompt and output for chain-of-thought question answering. The linearization consists of the question, the passage, and the final line "Explanation: ". The language model then generates the explanation and answer.

# B  Additional evaluations

**Entity-swap perturbation.**    Table 3 shows the results of a stress test evaluation that tests dependence on knowledge acquired during pretraining. Similar to [19], we perturb existing SQuAD examples by running a named entity recognizer and replacing names that appear in the answer and passage with names of other entities of the same broad class (e.g., "Winston Churchill" → "Patti Smith", "AT&T" → "the Denver Broncos.") The perturbations are performed only on the evaluation data, so we are evaluating the ability of a model fine-tuned on the original SQuAD data to generalize to these perturbations. Note that in some cases these perturbations affect the grammaticality of the passage, making the task more difficult for reasons that do not relate to the fidelity of the explanations. As shown in the table, all models are approximately 3-4 $F_1$ points worse than on the original evaluation set, with comparable exact match. This suggests that the predictors mainly relied on the passage and not on knowledge obtained during pretraining.

**Decontextualization.**    Detailed results from the evaluation on labeled decontextualizations [4] are shown in Table 4.

|  | em / $F_1$ |
| --- | --- |
| End-to-end | 83.7 / 89.3 |
| Markup+mask | 81.5 / 87.4 |
| Mask-only | 81.5 / 87.0 |

Table 3: Performance of the XXL-based student model on the SQuAD challenge set with entity perturbations.

|  | $F_1$ | Precision | Recall |
| --- | --- | --- | --- |
| **Students** | | | |
| XXL/QuoRef | 0.33 | 0.67 | 0.22 |
| XXL/SQuAD | 0.32 | 0.65 | 0.21 |
| **Teachers** | | | |
| 540B | 0.32 | 0.62 | 0.21 |
| 64B | 0.22 | 0.49 | 0.15 |
| 8B | 0.11 | 0.40 | 0.06 |
| **Fine-tuned** [4] | | | |
| T5-Base | 0.29 | 0.67 | 0.19 |
| T5-XXL | 0.42 | 0.72 | 0.30 |

Table 4: SARI-add metrics for decontextualization on the test set of [4]. The student models are distinguished by the behavior cloning dataset, which contains the answers but no labeled decontextualizations. Smaller student models performed almost identically to the XXL-scale models on this metric, but as shown in the table, smaller teachers were significantly worse.

## C   Implementation details

**Teacher model decontextualization.**   Sentence-level decontextualization requires sentence segmentation, which was performed using `sent_tokenize` function of NLTK [1]. Because sentence tokenization errors frequently propagated to decontextualization errors, we applied a few hand-crafted character-level replacement rules to improve segmentation accuracy, e.g. transforming expressions like *J. R. R. Tolkien* into *J.~R.~R. Tolkien*. All such transformations were reversed after sentence segmentation. The maximum number of context sentences was set at $k = 5$.

## D   Error analysis

On both datasets, the biggest source of erroneous answers for the pipeline model was the selection of rationales that do not contain the gold answer. In QuoRef, many questions are multihop, requiring information found in multiple spans in the passage. In some cases this information can be localized by the markup — as in the motivating example shown in Figure 1. There were several reasons that markup failed add the information necessary to provide a localized rationale:

- Sometimes, the necessary markup could have been supplied but was erroneously omitted: for example, to the question *who is Fran's son?*, the pipeline model provides the rationale *The spirit reminds Scrooge [Ebenezer Scrooge] that Fran, dead for some years, is the mother of his [Ebenezer Scrooge's] nephew*, which would have been sufficient if additional markup had been provided after the word *nephew*.

- On the QuoRef dataset, a large class of errors relates to markup that was supplied for names. Many of the questions involve nicknames and pseudonyms, and the markup sometimes included the wrong name, which then propagated to the reading comprehension module. In other cases, part of the name was lost, such as the disappearance of the given name of *Diego Velázquez* in the markup in Figure 6.

- Implicit entity references are not disambiguated by markup: for example, the sentence *In 1905 Ravel, by now thirty, competed for the last time, causing a furore* introduces a piano competition, which would have to be disambiguted for the sentence to serve as a rationale for the question *What is the name of the competition Ravel entered for the last time in 1905, inadvertently causing a furore?*

- Some questions reference multiple facts in the passage, such that it is difficult to imagine any markup making it possible to localize a rationale into a single sentence. For example, for the question *in what country did Rakoto Frah's troupe win the gold medal?*, the selected rationale is *Among the 80 competitors hailing from a variety of countries, Rakoto Frah's [the artist] troupe won the gold medal*, which is the only sentence mentioning the event from the

| Dataset | Rationale | e2e == pipeline? | Coverage | EM | F1 |
|---------|-----------|:----------------:|----------|------|------|
| **SQuAD** | **markup+mask** | ✓ | 86.8% | 88.0 | 95.3 |
| | | ✗ | 13.2% | 51.8 | 75.8 |
| | **mask-only** | ✓ | 87.4% | 87.7 | 95.1 |
| | | ✗ | 12.6% | 52.2 | 76.4 |
| **QuoRef** | **markup+mask** | ✓ | 74.2% | 88.0 | 91.5 |
| | | ✗ | 25.8% | 58.3 | 69.4 |
| | **mask-only** | ✓ | 57.5% | 87.3 | 91.3 |
| | | ✗ | 42.5% | 70.9 | 78.3 |

Table 5: Evaluation of selective prediction for the XXL-based models. Answers from the end-to-end predictor are distinguished by whether they agree with the answer provided by the honest student pipeline. For example, the top row shows that on SQuAD, the predictors agree on 86.8% of examples, receiving an $F_1$ of 95.3 on this subset.

question. To provide the answer, the markup would have had to supply location information for the event *won the gold medal*. If this was done as a general practice, significantly more markup would have been required.

- Finally, in some cases the rationale selector simply failed to select a rationale that answered the question. For example, given the SQuAD question *which entity has a monopoly on initiating legislation?*, the pipeline model selected the rationale *It [The Parliament of the European Union] can require the Commission [of the European Union] respond to questions and by a two-thirds majority can censure the Commission [of the European Union]*, missing the better rationale *the Commission has a monopoly on initiating legislation*.

In general, when the rationale did not contain sufficient information to answer the question, the pipeline model "hallucinated" the requested details. However, as shown in Section 4.2, this was not typical: on both datasets, the rationales usually entail the predicted answer.

# E   Selective prediction results

Table 5 shows the results for selective prediction, distinguishing cases in which the end-to-end answer matches the pipeline from cases where they do not match. When the two answers do not match, the end-to-end system is evaluated because it is more accurate overall.