# OpenReview forum: "Honest Students from Untrusted Teachers: Learning an Interpretable Question-Answering Pipeline from a Pretrained Language Model"
_NeurIPS.cc/2022/Workshop/TSRML — TSRML2022_

### Official Review · Reviewer_CEnj · 2022-10-19
**A solid work proposing a new type of extractive rationales and an effective knowledge distillation paradigm**

**Overall Rating:** 9

**Summary:**

The major contributions of the work are two-fold. Firstly, the work proposes a new type of extractive rationales, markup-and-mask, which are grounded by the original context and yet can be read alone. Secondly, the work proposes to use a prompted language model for obtaining silver training data, which is later used to train a student model. The student model is designed as a pipeline, making sure the answer is derived from the rationales. Experiments demonstrate the effectiveness of the proposed method and a list of comprehensive analyses also validates the source of improvement.

**Strengths:**

1. The proposed design of markup-and-mask rationales is sound and novel. It's also proved to be useful as it increases the fraction of rationalizable answers.
2. The work adopts a prompted LM for obtaining silver training data for fine-tuning a student model, which is more efficient compared to human annotation. It also adopts some validations to ensure the quality of the data. The design of the student model is also effective in making the student model faithful to its generated rationales.
3. The experiments are comprehensive, providing different perspectives to understand how different parts of the whole system contribute to the final performance.

**Weaknesses:**

1. The work does not verify the training data for decontextualization as it does for rationale generation. Yet, the experiment shows that the student model fine-tuned in this way performs better than the teacher model. It would be helpful if the authors can have a discussion in the paper about what new supervision signals are brought during the process of the distillation.
2. One baseline that should be considered is to directly train the student model for generating rationales using the heuristic-based extracted rationales.



**Overall Recommendation:**

Overall I recommend accepting the paper given the novelty in terms of a new type of extractive rationales and an effective knowledge distillation paradigm. The results are all convincing given the thorough experiments.

**Review Confidence:**

4: The reviewer is confident but not absolutely certain that the evaluation is correct

---

### Official Review · Reviewer_p2cT · 2022-10-21

**Overall Rating:** 6

**Summary:**

In this paper, a markup-and-mask rationalization method is proposed for open-book question answering. For student model, to answer a question, an augmented passage with free text markup are fed into span selector to identify rationales and only those selected rationales are utilized in reading comprehension. The main difference between the teacher model and the distilled student model is that in the student model rationale acts as a bottleneck between the passage and the answer. Prompt learning is incorporated to generate silver annotations.

**Strengths:**

- The description of annotation generation is very detailed, also utilizing the combination of  in-context learning and end-task supervision to avoid the cost of gathering human annotations sounds like an interesting idea.

- Extensive experiments are provided and results are impressive, which show the usefulness of the framework. Also, current limitations are discussed in detail.


**Weaknesses:**

- The paper is not easy to follow because there is no smooth connection between different chunks. It’s better to write a pipeline figure to show how the whole framework works.

- One weakness of the paper is that it doesn't provide a math formula for their selective rationalization, which includes the usual sparsity constraint and coherent constraint. It is better to show how they optimize the objective function with hyper-parameters related to rationalization.


**Overall Recommendation:**

This paper delivered an interesting idea with extensive experiments, but the writing flow is not smooth, sometimes hard to follow.

**Review Confidence:**

3: The reviewer is fairly confident that the evaluation is correct

---

### Decision · Program_Chairs · 2022-10-23

**Decision:**

Accept

**Comment:**

Following the unanimous recommendations from reviewers, the submission is accepted.